# The Effects of Cannabinoids on Executive Functions: Evidence from Cannabis and Synthetic Cannabinoids—A Systematic Review

**DOI:** 10.3390/brainsci8030040

**Published:** 2018-02-27

**Authors:** Koby Cohen, Aviv Weinstein

**Affiliations:** Department of Behavioral Science, Ariel University, Ariel 40700, Israel; kbbcohen@gmail.com

**Keywords:** cannabis, synthetic cannabinoids, executive function

## Abstract

Background—Cannabis is the most popular illicit drug in the Western world. Repeated cannabis use has been associated with short and long-term range of adverse effects. Recently, new types of designer-drugs containing synthetic cannabinoids have been widespread. These synthetic cannabinoid drugs are associated with undesired adverse effects similar to those seen with cannabis use, yet, in more severe and long-lasting forms. Method—A literature search was conducted using electronic bibliographic databases up to 31 December 2017. Specific search strategies were employed using multiple keywords (e.g., “synthetic cannabinoids AND cognition,” “cannabis AND cognition” and “cannabinoids AND cognition”). Results—The search has yielded 160 eligible studies including 37 preclinical studies (5 attention, 25 short-term memory, 7 cognitive flexibility) and 44 human studies (16 attention, 15 working memory, 13 cognitive flexibility). Both pre-clinical and clinical studies demonstrated an association between synthetic cannabinoids and executive-function impairment either after acute or repeated consumptions. These deficits differ in severity depending on several factors including the type of drug, dose of use, quantity, age of onset and duration of use. Conclusions—Understanding the nature of the impaired executive function following consumption of synthetic cannabinoids is crucial in view of the increasing use of these drugs.

## 1. Introduction

The most popular illicit drug of the 21st century is cannabis, in its many forms and shapes [1,2,3,4,5]. According to the United Nation Office on Drugs and Crime (UNODC), approximately 181 million adults have used cannabis across the globe [2]. Moreover, in many countries more than 50% of young adults have used cannabis at least once in their lifetime [3]. Recently, new types of drugs that contain Synthetic Cannabinoids (SC) have become popular among drugs users worldwide [5,6,7]. SC drugs are associated with severe adverse effects (Table 1), have greater harm potential and they are more addictive than the traditional organic cannabis drugs [4,6,7,8,9]. Therefore, governments and health institutions across the Western world make major efforts in order to prevent the spread of SC and to improve the knowledge regarding SC and their potential risks [5,8]. One of the most notorious adverse effects that is associated with cannabinoids consumption is impairment of cognitive function [4]. Both pre-clinical and human studies drew a link between consumption of cannabinoids and long-term deficits of cognitive functions, especially high-order cognitive functions [4,5,10,11,12,13]. The purpose of the current review is to present and describe the acute and long-term effects of SC drugs in comparison with organic cannabis on executive function (EF) based on current literature from both human and animal research. A literature search was conducted using electronic bibliographic databases (PubMed®, ScienceDirect ® and Google Scholar platforms) up to 31 December 2017. Database-specific search strategies were employed using multiple keywords (e.g., “synthetic cannabinoids AND cognition,” “cannabis AND cognition” and “cannabinoids AND cognition”). The search has yielded 160 eligible studies including 37 preclinical studies (5 attention, 25 short-term memory, 7 cognitive flexibility) (Table 2) and 44 human studies (16 attention, 15 working memory, 13 cognitive flexibility). Studies were included if they stated the following inclusion criteria: use of objective measurements of specific executive function (working memory, attention and cognitive flexibility) involving cannabinoid users (regular and recreational users) or cannabinoid treatments and a control group. Exclusion criteria were: studies that involved participants who had other neurological or psychiatric disorders or individuals who met criteria for alcohol dependence or other substance use disorders (abuse or dependence) different from cannabis and nicotine.

## 2. Pharmacology of Organic Cannabis

Cannabis is the generic name of the psychoactive drug that is derived from the female plant *Cannabis sativa* [51]. There are more than 400 compounds including more than 60 cannabinoids, which are aryl-substituted meroterpenes unique to *Cannabis sativa* [52,53]. The main psychoactive ingredient in cannabis is Δ9-Tetrahydrocannabinol (Δ9-THC), which is the most potent cannabinoid that is present in the organic form of cannabis [53]. Besides Δ9-THC, organic cannabis products contain additional cannabinoids which do not induce psychoactive effects, such as Cannabidiol (CBD), Δ8-Tetrahydrocannabinol and Cannabinol [54,55,56]. Furthermore, CBD is considered a non-psychoactive cannabinoid that also moderates the psycho tropic effects of Δ9-THC [57,58,59,60]. 

The psychoactive effects of cannabis are dose-dependent [58,61,62] and there is evidence that as the content of Δ9-THC increases, the psychoactive effects of cannabis drugs increase [59,62]. Cannabinoid agonists in general and specifically Δ9-THC, exert their effects by acting on at least two types of endogenous cannabinoid receptors (CB1, CB2), which are widely distributed in numerous regions within mammals’ brain [52,63,64]. Pacher and Kunos (2013) suggested that endocannabinoid receptors, the two endocannabinoid ligands and their related enzymes are the components of the Endo-Cannabinoid System (ECS), which is involved in a wide range of somatic and mental functions [65]. 

## 3. Synthetic Cannabinoids, from Therapeutic Agents to a Global Disease

### 3.1. Old Origins, New Trends

Since the discovery of Δ9-THC and the involvement of the ECS in a wide range of health conditions, cannabinoids have been synthesized for medical research purposes as promising research and therapeutic tools [24,23]. In contrast to organic cannabinoids such as Δ9-THC, SCs selectively activate the endocannabinoid receptors [24,56,57]. 

In the beginning of the new millennium, a growing number of reports indicated that there were new psychoactive products which included mostly SC ingredients mixed with other herbal blends [6,66,67]. The production, distribution and use of SC drugs were initially neither controlled nor illegal, therefore they are presented as “legal-highs” [67], by various generic names such as; “Mr. Nice Guy,” “Spice Gold,” “Spice Diamond,” “Yucatan Fire” and most commonly as “K2” or “Spice” [7,25]. These products were often sold without age restriction over multiple sources such as the internet and convenience stores [4,7,8,9,25,67]. As the popularity of SC drugs increased, their severe undesired adverse effects were observed; affective disorder, recurrent psychosis, tachycardia, seizures and prolonged hospitalization were not rare outcomes of SC intoxication [4,5,7,8,9]. Some of these adverse effects are related to the effect of additional psychoactive agents which these products contain [6,8,68].

Despite the fact that SCs are labeled as “not for human consumption” and “for aroma therapy use only,” the popularity of these drugs appears to be growing [5]. SCs induce more intense effects than traditional cannabis, they are less expensive and they are undetectable in standardized drug tests. These unique features contribute to the growing numbers of recreational drug users who have used SCs [4,5]. 

### 3.2. The Psychoactive Ingredients of Synthetic Cannabinoid Products

Over than 140 products containing SC have been identified, although, the main psychoactive components of these products are different types of SCs which are categorized into four major groups including; (a) Aminoalkylindole or JWH series, (b) classical cannabinoids, (c) non-classical cannabinoids and (d) fatty acid amides (e.g., oleamide) [21,22,69,70].

The first generation of SC products mostly contain the series of 1-alkyl-3-(1-naphthoyl) indoles known as JWH compounds or aminoalkylinodels. This SC series is named after John W. Huffman who developed these ligands for medical research purposes [71] The JWH series advanced from computational melding of the chemical structural structures of Δ9-THC with previously developed aminoalkylinodels [71]. One of the first SC from this series to be abused is JWH-18 (1-penthyl-3-(1-napthoyl) indole), which features as easy synthesizable and high potency contribute to its popularity [9]. Compared to Δ9-THC, JWH-018 has 4 times the affinity for CB1 receptors and 10 time the affinity for the CB2 receptors [72]. JWH series represent the main psychoactive compounds detected in SC products across many countries [9]. 

Additional components detected in SC products include analogues of Δ9-THC, so-called classical cannabinoids such as HU-210 and HU-211. HU-210 developed in the middle of the 20th century at the Hebrew University (HU) [73] and is a hundred times more potent than Δ9-THC binds both CB_1_ and CB_2_ receptors [73,74] Similar to other SC, HU-210 acts as CB1 receptors full-agonists [73]. 

The cyclohexylphenol (CP) is a non-classical cannabinoids series synthesized by *Pfizer labs* in the early 1970s; examples include CP 59,540, CP 47,497 and their n-alkyl homologues [71]. Similar to JWH-018, CP-47,497 is included in large numbers within SC products e [67]. In addition, SCs from the CP series act as CB1 receptors full agonists [67]. However, within any given SC products, various types of SC are found in different concentrations [9,67] accompanied by additional psychoactive compounds from synthetic opioids such as O-desmethyltramadol, harmine and harmaline, which are inhibitors of the monoamine oxidase enzyme, to benzophenone (HM-40) and even caffeine [9,11,68]. There are several common features among different compounds of SC products which can highlight the risk potential which these drugs have and their related adverse effects. Firstly, SCs act as full agonists to CB1 receptors and some also bind to CB2 receptors [7]. Secondly, SCs are much more potent, easily cross the blood-brain barrier and have more affinity compared to organic psychoactive cannabinoids like Δ9-THC [68,69]. In addition, SC drugs do not contain CBD, which has high potency as an antagonist to CB1 and CB2 receptors and therefore it is able to revert the psychotic and anxiolytic adverse effects of cannabinoid-agonists. It is suggested that the lack of CBD in SC products amplifies their psychotropic effects [4,6,75]. Moreover, SC products hold a unique characteristic, which is its ever-changing composition. The first generation of SC products commonly contain JWH-018, JWH-073 and CP-47,49, since these SCs became regulated, there has been an emergence of new types of SCs like JWH-081, JWH-210 and AM-2201, in an attempt to dodge regulations. Despite slight chemical structure modification, all of these SCs share the same main features and aim to mimic the psychoactive effects of Δ9-THC and even to transcend it [4,6,66,67,75]. 

The cyclohexylphenol (CP) is a non-classical cannabinoids series synthesized by *Pfizer labs* in the early 1970s; examples include CP 59,540, CP 47,497 and their n-alkyl homologues [71]. Similar to JWH-018, CP-47,497 is included in large numbers within SC products e [67]. In addition, SCs from the CP series act as CB1 receptors full agonists [67]. However, within any given SC products, various types of SC are found in different concentrations [9,67] accompanied by additional psychoactive compounds from synthetic opioids such as O-desmethyltramadol, harmine and harmaline, which are inhibitors of the monoamine oxidase enzyme, to benzophenone (HM-40) and even caffeine [9,11,68]. There are several common features among different compounds of SC products which can highlight the risk potential which these drugs have and their related adverse effects. Firstly, SCs act as full agonists to CB1 receptors and some also bind to CB2 receptors [7]. Secondly, SCs are much more potent, easily cross the blood-brain barrier and have more affinity compared to organic psychoactive cannabinoids like Δ9-THC [68,69]. In addition, SC drugs do not contain CBD, which has high potency as an antagonist to CB1 and CB2 receptors and therefore it is able to revert the psychotic and anxiolytic adverse effects of cannabinoid-agonists. It is suggested that the lack of CBD in SC products amplifies their psychotropic effects [4,6,75]. Moreover, SC products hold a unique characteristic, which is its ever-changing composition. The first generation of SC products commonly contain JWH-018, JWH-073 and CP-47,49, since these SCs became regulated, there has been an emergence of new types of SCs like JWH-081, JWH-210 and AM-2201, in an attempt to dodge regulations. Despite slight chemical structure modification, all of these SCs share the same main features and aim to mimic the psychoactive effects of Δ9-THC and even to transcend it [4,6,66,67,75]. 

The cyclohexylphenol (CP) is a non-classical cannabinoids series synthesized by *Pfizer labs* in the early 1970s; examples include CP 59,540, CP 47,497 and their n-alkyl homologues [71]. Similar to JWH-018, CP-47,497 is included in large numbers within SC products e [67]. In addition, SCs from the CP series act as CB1 receptors full agonists [67]. However, within any given SC products, various types of SC are found in different concentrations [9,67] accompanied by additional psychoactive compounds from synthetic opioids such as O-desmethyltramadol, harmine and harmaline, which are inhibitors of the monoamine oxidase enzyme, to benzophenone (HM-40) and even caffeine [9,11,68]. There are several common features among different compounds of SC products which can highlight the risk potential which these drugs have and their related adverse effects. Firstly, SCs act as full agonists to CB1 receptors and some also bind to CB2 receptors [7]. Secondly, SCs are much more potent, easily cross the blood-brain barrier and have more affinity compared to organic psychoactive cannabinoids like Δ9-THC [68,69]. In addition, SC drugs do not contain CBD, which has high potency as an antagonist to CB1 and CB2 receptors and therefore it is able to revert the psychotic and anxiolytic adverse effects of cannabinoid-agonists. It is suggested that the lack of CBD in SC products amplifies their psychotropic effects [4,6,75]. Moreover, SC products hold a unique characteristic, which is its ever-changing composition. The first generation of SC products commonly contain JWH-018, JWH-073 and CP-47,49, since these SCs became regulated, there has been an emergence of new types of SCs like JWH-081, JWH-210 and AM-2201, in an attempt to dodge regulations. Despite slight chemical structure modification, all of these SCs share the same main features and aim to mimic the psychoactive effects of Δ9-THC and even to transcend it [4,6,66,67,75]. 

## 4. Executive Function (EF) and the Long-Term Effects of Cannabinoids

### 4.1. The Three Core Factors Model of Executive Function

Although preclinical and human studies demonstrate that endocannabinoids involve and affect cognitive function in general and specifically high-order cognitive function [12,13,51,68], there is still a debate regarding the effects of chronic consumption of cannabinoid products such as cannabis or SCs on EF [12,13,46,70] (Table 1).

The term EF refers to “high-order” cognitive functions, which involve regulation, “lower-order” cognitive process and goal-directed behaviors [76,77]. EF generally clusters various cognitive abilities such as verbal reasoning, problem-solving, planning behaviors, sequencing, multi-tasking, cognitive flexibility, sustained attention, resistance to interferences and the ability to deal with novel information [77,78,79,80]. Due to the wide range of functions which are considered as executive or high-order, there is still an ongoing debate regarding the mechanisms which underlie executive function, performances and regarding which cognitive functions should be marked as executive [76].

Diamond (2013) suggested that EF should be divided into two subgroups: core EF and higher order EF [77]. Accordingly, the three cores EFs are (a) inhibition control or attention (b) Working Memory (WM) and (c) cognitive flexibility. The basic EFs are essential for the production of higher order cognitive functions such as verbal reasoning, problem-solving, planning behaviors, sequencing and multi-tasking. Accordingly, these functions do not involve much emotional arousal and they are logic based [77]. 

### 4.2. Cannabinoids and Attention-Evidence from Preclinical Studies

The ability to evaluate and allocate priority to external stimuli or internal habits and to optimize behavioral response requires attention [13,77]. These enable focus and selectively attend to desired stimuli and to inhibit response to irrelevant stimuli [77]. Studies have suggested that numerous brain regions facilitate attention performance, yet, it is mediated by the frontal lobes [81,82]. Additionally, the Anterior Cingulate Cortex (ACC) is a crucial factor in the execution of this function [82,83].

Preclinical studies provide strong evidence regarding the effects of repeated treatment with cannabinoid-agonists and impaired attention. The Lateralized Reaction Time task (LRT) of visuo-spatial attention that has been previously used in rats, is considered as a valid model for attention in rodents. In this paradigm, rodents need to attend to apparatus for the location of a visual stimulus over numbers of trails [83]. Arguello and Jentsch (2004) reported that acute treatments with the SC agonist WIN55212-2 (2.5 mg/kg) induced deficits in attention measured on the LRT task. In addition, treatment with SR141716A 1 mg/kg which is a CB1 antagonist reversed the WIN55212-2-induced attention impairment, although, when administered alone, this compound did not produce any effects on attention [26].

A further study by Verrico et al (2004) examined the effect of repeated treatments with Δ9-THC on attention using the LRT task in rats. In their study, rats that were daily treated with Δ9-THC 20 mg/kg for 2 weeks, presented attention impairments which lasted 14 days after the last treatment with Δ9-THC [28]. Later-on, Miller et al. (2013) treated rats with small doses of novel SC agonists AM-4054 before performing a two-choice reaction time task, which measures sustained attention. They reported that AM-4054 induced attention impairments which were positively correlated with task demands and harder trails were associated with poorer functions [14]. 

Some authors suggested that lesions of the medial prefrontal cortex or striatum can produce attention deficits similar to those presented after cannabinoid administration [84,85]. Chronic exposure to cannabinoid-agonists led to alterations within meso-limbic dopaminergic neurons [86], thus, cannabinoid-induced attention impairment might arise via continuous activation of CB1 receptors across the striatum or prefrontal cortex [83]. 

### 4.3. Cannabinoids and Attention-Evidence from Clinical Studies

The disruptive acute effect of cannabis on attention is widely described in clinical studies [87,88,89,90] and systematic reviews [11,70,87]. Yet, human studies failed to draw consistent evidence regarding the effect of chronic consumption of cannabinoids and impaired attention. While some studies described impairments of tasks which demand attention in chronic cannabis users [17,19,61,89,91], other studies demonstrated no differences in behavioral performance between cannabis users and non-users [22,87]. Since neuronal and functional alterations of the ACC region were consistency observed among chronic cannabis users [92] a recent review study suggested that the marginal effects that were observed in these studies are probably an outcome of a compensation mechanism that was developed among chronic users [87]. 

There are several tasks for measuring attention. In a paradigm such the Stroop task, a control of interference from of a pre-potent response is required [93]. Incongruent conditions of the classical Stroop color-word task contain color words written in another color. Subjects are required to ignore the semantic meaning of the word and instead attend to and report the color. Since humans are trained to read and to ignore other words’ features such as font style or color, people are slower and prone to make more errors in the incongruent trials of the Stroop task [77]. 

On the Go/No-Go task, the participants do not inhibit natural response at the expense of another. On this task, participants are required to respond when target stimulus is presented and should not respond when a non-target stimulus appears [94]. Other tasks such as the Continuous Performance Task (CPT) are being used for measuring sustained attention. In this paradigm, participants are required to maintain attention over a continuous period in order to detect infrequent targets, thus ensuring that the goals of the behavior are kept over time [20].

Eldreth and colleagues (2004) have examined the performance on a modified Stroop task in which healthy individuals were compared with abstinent cannabis users. Although there were no behavioral differences between the groups, cannabis users had greater activation in prefrontal brain regions than non-users [95]. Similarly, Jager et al. (2006) observed moderate differences in brain activity between cannabis users and healthy individuals while performing attention and WM tasks. They reported that compared with healthy subjects, cannabis users presented hypo-activation in the left superior parietal cortex while performing the attention task [96]. 

Recently, Hatchard and colleagues (2014) observed a similar pattern among young cannabis users. Recreational cannabis users did not differ in performance on the modified Stroop task compared with non-users, however, differences in neuronal activity of several brain regions including the ACC and post-central gyrus were observed, suggesting that chronic consumption of cannabis affects neuronal process even in an absence of behavioral expressions [97]. In another study, Hester et al (2009) reported that alterations in attention correlated with neuronal hypo-activity of ACC in heavy cannabis users. The attention deficits expressed in performing more errors on the Go/No-go task, suggested that attention depended on cannabis consumption history, including doses, frequency and age of onset [98]. 

The studies described so far examined the complex association between chronic consumption of organic cannabis and impaired attention, yet, there is limited objective evidence for an association between chronic consumption of SCs and impaired attention in humans [9]. Cohen et al (2017) showed that SC users had more errors performing on the classic Stroop color-word task compared with regular cannabis users and healthy subjects [11]. Furthermore, several case reports described SC users who experienced “thinking problems” which last from days to weeks following last consumption. However, attention deficits were less common and they were accompanied with additional symptoms such as affective disturbances and cognitive dysfunction including severe alterations in short-term memory [8,99,100]. 

### 4.4. Cannabinoids and Working Memory-Evidence from Preclinical Studies

Working Memory (WM) is defined as a cognitive mechanism for the temporary storage and manipulation of stored information [101], or simply, as a cognitive system which involves holding information in mind and mentally working with it [77]. 

The function of WM has been associated with integration of a wide range of neural networks. WM networks are associated with frontal-parietal regions including dorso-lateral pre-frontal cortex, ventro-lateral prefrontal cortex, pre-motor cortex, lateral parietal cortex and the frontal lobe [102]. An additional brain region which is considered a major component in WM is the hippocampus, which is essential for acquiring, encoding and consolidating new types of information. This information is represented and manipulated by the WM system in the prefrontal cortex [103]. In rodent models, changes in hippocampal morphology were observed following chronic treatments of various doses with cannabinoid agonists like Δ9-THC and WIN55,212-2, these neuronal alterations correlated with behavioral dysfunction [30,31,32].

Preclinical studies which used rodent as animal models, utilized both maze-based and instrumental tasks for investigating the effect of cannabinoids on WM [31]. Maze-based tasks require the rodent to use spatial cues correctly. These tasks are based on the navigational behaviors of rodent for foraging or in order to escape from predators [104]. Several works have suggested that chronic treatment of Δ9-THC induced WM impairments on in different types of maze-based tasks [33,34] and in water maze tasks [35,36,38]. These impairments are dose-related, thus greater impairments were observed after exposures to more potent cannabinoid-agonists [36,41]. Therefore, it is not surprising that SC agonists such as JWH-081 and HU-210 induce similar disruptive effect on WM performances in maze-based tasks [37,39]. In addition, similar impairment is induced with anti-cholinergic agents like physostigmine, suggesting that cannabinoid-agonists induce WM impairments due to interaction with acetylcholine system [105].

Instrumental WM tasks in rodents include the delayed matching to sample (DMTS) or delayed non-matching to sample (DNMS) tasks. During these tasks, the animal is initially presented with a stimulus and following delay period, both the original stimulus and a novel stimulus are presented. The animal must indicate either the sample stimulus or the novel stimulus follow the task’s rule [31]. The effects of chronic treatment with cannabinoid agonists such as WIN55212-2 and Δ9-THC on WM in DMTS or DNMS paradigms are widely observed, both in rodents [27,41,42,106] and in primate models [107,108]. Again, most of the studies report that the disruptive effects of cannabinoid agonists are dose-dependent [31]. 

Recently, Barbieri et al. (2016) reported that administration of a CB1 receptors antagonist AM251 to mice as pre-treatment, fully prevented the disruptive effects of cannabinoid agonists including JWH-018 and Δ9-THC on WM, thus suggesting a CB1 receptor involvement in the effect of cannabinoids on WM [43]. Other studies reported that repeated treatment with SC agonists JWH-018 and CP55,940 in the puberty period induced severe WM impairments that remained in adulthood [44,45]. These findings are consistent with previous theories which suggested the involvement of ECS in brain development and that consumption of cannabinoid agonists in adolescence alter the function of the ECS [24,32,109,110]. 

Interestingly, some studies report contrary results where reduced impairments following repeated treatment with cannabinoid agonists were presented [38], although, this might be a result of tolerance [31]. In addition, further preclinical research is needed to examine the degree of persistence of deficiencies induced by chronic treatments with cannabinoid agonists [31]. Yet, a growing number of publications indicate that exposures to cannabinoids in early age are associated with greater and persistent WM deficits, suggesting that the age of onset may be a mediating factor in the association between cannabinoids and WM performance [32,106,111,112,113]. 

### 4.5. Cannabinoids and Working Memory- Evidence from Clinical Studies

The disrupted effect of acute cannabis intoxication on WM performance in humans is widely documented [87,33]; however, there is a growing debate whether chronic cannabinoid consumption induces long-term impairments of WM [87,114,115].

The most common paradigm for measuring WM performance is the n-back task. During this task a sequence of constant stimuli in form of digits, shapes or numbers are presented to the subject, who need to decide if the presented stimulus is identical to a previous stimulus from n steps earlier. The load factor n reflects different WM loads; lower n represents an easier task [116]. Kanayama and colleagues (2004) investigated WM in chronic cannabis users and used functional Magnetic Resonance Imaging (fMRI) [117]. They reported that cannabis users did not show WM dysfunction; however, increased activation of several brain regions including prefrontal-cortex, ACC and basal-ganglia regions were observed. The authors suggested that chronic consumption of cannabis induced subtle neurophysiological deficits which are compensated by hyper-activation to meet the demands of the task [117,118].

In addition, an fMRI study which focused on hippocampus activity during performance on the n-back task, compared cannabis users with two control groups of healthy individuals and tobacco smokers [119]. Poorer performance was observed in cannabis users compared with both control groups on the task’s overall score. Furthermore, cannabis users presented less activity in the right hippocampus across the task’s conditions contrary to both control groups [119]. In a further neuroimaging study, Jager et al. (2007) examined the effects of cannabis use on neuronal activity in abstained cannabis users and healthy control participants during performance on the n-back task consisting of encoding and recall conditions [120]. Similar to previous studies [121], there were no differences between the groups in terms of behavioral performance. Interestingly, cannabis users exhibited hypo-activation in the right dorso-lateral pre-frontal cortex and in bilateral hippocampus regions. This reduced activity in WM responsible areas were limited to the encoding phase and were not presented in the rest of the task phases’ [120]. 

Smith and colleagues (2010) used fMRI to examine the neuronal brain activity of heavy cannabis users and control non-users while performing different loads of the n-back task. The two groups presented similar WM performance, however, in contrast to other studies, cannabis users demonstrated hyper-activity in the right frontal gyrus, left middle inferior frontal gyrus and right superior temporal gyrus [122]. In a recent systematic review, Bossong and colleagues (2014) suggested that most functional neuroimaging studies present similar pattern of hyper neuronal brain activity in cannabis users compared with control participants that were accompanied with normal WM function [123]. They support the view that increased activity reflects greater neural effort in order to maintain good task performance [123]. On the other hand, a-3-year longitudinal neuroimaging study failed to find behavioral or functional differences between cannabis users and control participants, suggesting that a moderate use of cannabis may not have substantial effects on WM neural network and behavioral performance [124]. However, WM deficits in chronic cannabis users are more likely to be elicited in complex conditions [115]. Therefore, a lack of differences in WM performance between cannabis users and control participants does not necessarily indicate a lack of association between chronic consumption of cannabis and WM [123]. 

Convergent evidence from structural neuroimaging studies supports the last view indicating that chronic consumption of cannabis is associated with neuronal alterations in several brain regions which are involved in WM including reduction in size of the hippocampus and amygdala. In addition, these alterations correlated with the amount of cannabis use and dependence [125]. Recently, Battistella et al. (2014) reported similar data, where neuronal alterations in several brain regions including the parahippocampal gyrus were observed in chronic cannabis users compared with occasional users. Furthermore, these alterations are associated with age of onset and frequency of cannabis use in the last 3 months [126]. 

To our knowledge, there is a limited number of available laboratory human studies investigating the association between persistence consumption of SC with WM performance. Yet, Castellanos and Thornton (2011) reported that young adults who used SC drugs experienced alterations in short-term memory; however, their main symptom was a severe psychotic episode [127]. Further reports described similar clinical manifestations where SC users experienced symptoms including alterations in short-term memory [128,129]. Cohen et al. (2017) demonstrated WM impairments observed among SC users compared with non-users and recreational cannabis users [11]. These reports are not surprising since CB1 receptors are highly distributed in the hippocampus and in prefrontal cortical regions [130,131], which are associated with WM [102]. In addition, SC products contain high-potency cannabinoid agonists, therefore it is reasonable that chronic consumption of SC induces impairments in WM function in more salient forms than those which are induced by organic cannabis [66,67]. 

### 4.6. Cannabinoids and Cognitive Flexibility- Evidence from Preclinical Studies

Cognitive flexibility has been described as the cognitive ability to think about multiple concepts simultaneously and to be able to switch between thinking about two different concepts [18]. Miyake et al. (2000) identified cognitive flexibility as the ability to shift one’s thinking and attention between unrelated tasks, typically in response to a change in environmental demands [81]. Diamond (2013) expanded the view of the term and suggested that an additional feature of cognitive flexibility is being able to change perspectives spatially or inter-personally. Accordingly, for changing perspectives, an individual needs to inhibit the last perspective and to load a new perspective into WM [77]. In that sense, cognitive flexibility builds and depends on WM and inhibition control. Other aspects of cognitive flexibility involve changing the way of thinking in response to external demands and thinking “outside the box” [77].

In rodents, variations of attention set-shifting paradigms are being used to assess behavioral flexibility. During these tasks rats are required to change behavioral responses, by learning new stimulus-reward associations through earlier learned response inhibition tendencies [13]. These paradigms differentiate between two types of behavioral flexibility; (a) for successful extra-dimensional shifts the rats need to shift attention bias between different features of stimuli, (b) reversal-learning discriminations required the rats to update relations between stimuli and rewards presentation, in this inter-dimensional discrimination based on cue from a single modality [13,83]. This differentiation is important since these two aspects of behavioral flexibility are linked with different brain regions [13], while reversal learning is associated with orbito-frontal cortex [132], extra-dimensional shifts are mediated by the medial pre-frontal cortex [133,134].

Several preclinical studies investigated the effects of cannabinoid-agonists on cognitive flexibility and indicated inconsistent results. Egerton and colleagues (2005) reported that acute administration of 5 mg/kg Δ9-THC induced impairments in reversal learning, whilst attention set shifting ability was maintained [46]. Further primate research presented similar results using smaller doses and demonstrated that an acute administration of 0.5 mg/kg Δ9-THC induced more errors in reversal learning and it did not affect attention set shifting ability [135].

However, an additional rodent study has demonstrated different findings, whereby administration of 0.2 mg/kg of the SC agonists HU-210 2 days before measuring set-shifting, induced dose-dependent impairments in extra-dimensional set shifting ability [47]. These impairments were diminished after administration of a CB1 antagonist AM251. In addition, cannabinoids did not affect inter-dimensional reversal learning [47]. 

Further evidence regarding the effects of cannabinoids on cognitive flexibility was demonstrated by Varvel and Lichtman (2001). Knockout mice, which lack cannabinoid CB1 receptors presented impaired reversal learning in inter-dimensional water maze reversal learning task [40]. Their findings support the view that the ECS are involved in execution of cognitive flexibility [13]. Consistent with earlier studies, Gomes and colleges (2015) recently indicated that rats which were repeatedly treated with 1.2 mg/kg of a CB1 agonist WIN55,212-2 for 2 weeks in adolescence, showed deficits in adulthood in performance of set-shifting tasks and alterations in dopamine levels in the ventral tegmental area. These alterations were present in adulthood and were similar to those which were shown in pre-clinical models of schizophrenia [136]. 

The conflicting results demonstrated by previous studies, reflect the need for further studies on the effect of cannabinoids on cognitive flexibility. The available evidence demonstrates that cannabinoids have indeed an effect on cognitive flexibility [13], possibly via modulation of dopamine and glutamate concentrations in several brain regions including the ACC and prefrontal cortex [13,83]. 

### 4.7. Cannabinoids and Cognitive Flexibility- Evidence from Clinical Studies

Recent studies using fMRI have found a variety of brain regions that were activated while performing cognitive processes that demand flexibility, including, the pre-frontal cortex, basal ganglia, ACC and posterior parietal cortex [137]. Some of the regions which underlie cognitive flexibility are involved in WM and inhibition control and thus, the findings support the hypothesis that cognitive flexibility depends both on WM and inhibition control [81]. In addition, levels of certain neurotransmitters such as monoamines in several brain regions are associated with cognitive flexibility [138].

Paradigms for investigating cognitive flexibility include a wide array of task-switching and set-shifting tasks. One of the oldest and most common task for measuring this performance is the Wisconsin Card Sorting Task (WCST) [139]. In this task, a number of stimulus cards are sorted by color, shape or number. The participant is required to conclude the correct sorting criterion on the basis of feedback. Set-shifting ability is required when sorting criterion has been changed and perseverative errors are the outcome of failure in set-shifting [77]. Additional tasks for measuring cognitive flexibility include verbal fluency and semantic fluency. In these tasks participants are required to demonstrate unusual patterns of thinking by answering a serial of verbal questions (What is common between a fly and a tree?) in order to be successful [77]. 

Acute intoxication of cannabis has disruptive effects on cognitive flexibility [62,105,140]. However, the evidence on non-acute effects of cannabinoids on cognitive flexibility have been inconsistent. Bolla et al. (2002) reported dose-related effects of cannabis use on cognitive function. They have examined several aspects of cognitive function including cognitive flexibility in heavy cannabis users compared with moderate and occasional users who abstained from cannabis for 28 days. Poorer performance was positively correlated with increased frequency of cannabis consumption [61]. 

Later on, Pope et al. (2003) has reported similar effects, except that deficits in performance on the WCST were observed in heavy cannabis users who had started smoking cannabis during adolescence [141]. In addition, there were no differences in flexibility performance between cannabis users who had started using cannabis in adulthood compared with non-users [141]. A further study demonstrated that heavy cannabis users’ performance on the WCST resemble those of schizophrenic patients; however, there was no association between frequency of cannabis use and errors on the WCST [142]. Contrary to the last results, several studies indicated that while repeated consumption of cannabis has disruptive effects on some cognitive functions, impairments in cognitive flexibility were not presented in heavy cannabis users even after controlling for demographic variables [143,144]. 

In a systematic meta-analysis, Grant and colleagues (2003) examined the non-acute effects of cannabis on several aspects of cognitive function using strict inclusion criteria on a limited number of studies. The authors failed to find significant non-acute effects of cannabis consumption on cognitive flexibility. However, it should be noted that cognitive flexibility was referred as a component within the factor of abstraction reasoning [145]. This methodological issue is critical since abstract reasoning and cognitive flexibility are different components of EF [77].

The evidence so far points out to a lack of available evidence regarding the effects of SC on cognitive flexibility in humans. Altintas et al. (2016) examined several cognitive domains in SC users who experienced psychotic episodes and compared their performance with hospitalized schizophrenic patients. Interestingly, there were no differences between the groups in cognitive flexibility measurement [146]. Yet, their results cannot be interpreted as an outcome of SC use exclusively since it cannot be differentiated from psychotic symptoms that were observed among SC users as well. There are two additional aspects of the association between cannabinoid abuse and cognitive flexibility which should be noted. First, impairments in cognitive flexibility have been suggested to play a major role in continuous use of cannabinoids despite negative consequences [83]. Secondly, deficits in cognitive flexibility were associated with affective alterations [147]. Both greater mood alterations and greater rates of abuse are commonly observed among SC users and heavy cannabis users [9,148].

In summary, both pre-clinical and clinical findings suggest that the ECS are involved in cognitive flexibility [13]. Although, there are inconsistent findings in human studies, the non-acute disruptive effect of cannabinoids on cognitive flexibility is probably mediated by several factors including the age of onset and the frequency of cannabinoid consumption [58,141], yet, further exploration of the last relation is required. 

## 5. Conclusions

Cannabinoid drugs, in both organic and synthetic forms became increasingly popular despite the potential harms associated with their use [6,10,87]. While the main psychoactive ingredient of cannabis is the CB1 receptor partial-agonist Δ9-THC [13,51,52,53], SC drugs contain varied types of cannabinoid-agonists which are more potent than organic cannabinoids [65,149]. Although SC and organic cannabinoids bind to the same CB receptors, the psychotropic effects of SC are more severe, more rigid and much more unpredictable than those induced by organic cannabinoids [4,5,65,75]. Taking into account the above evidence that SC drugs do not contain CBD, their harm potential is significant [5,75,114]. 

Taking together the recent finding of both animal and human studies, repeated consumption of cannabinoids is associated with EF impairments, yet, there is still a gap of knowledge regarding the last of these impairments [11,114]. The available data from both animal and human studies suggest that ECS involve and effect cognitive functions in general and EF specifically [9,10,12,13,83]. The ECS has a major role in neurodevelopmental and maturational process, which are especially prevalent during adolescence. Consumption of exogenous cannabinoids affect the functioning of the ECS, it is plausible that chronic consumption during early adolescence alters the neurodevelopmental maturational process during this period [5]. Consequently, it is not surprising that current evidence suggests that exposure to cannabinoids during the adolescent period may induce severe long-lasting cognitive impairments [5,78,96,108,147,148]. Furthermore, most of the current evidence indicates an association between the amount of cannabinoid consumption with the degree of impairment; more consumption, or consumption of drugs which contain more potent cannabinoids is associated with greater impairments [5,83,87]. Accordingly, although there is a limited number of human studies which examine both the acute and long-term effect of SCs on EF, it is reasonable to assume that SC which contain extremely potent cannabinoid-agonists may induce long-term EF-impairments [5,6,7,8]. Yet, further research is needed to expend to knowledge of the last phenomena.

It is important to note some of the limitations of the current review. Most of the available evidence regarding the effects of SCs on EF is based on pre-clinical studies. When interpreting these results, it is important to take into account that the methodological limitations which animal studies naturally hold. Firstly, while cannabis or SC users mostly use these drugs by smoking or inhaling [1,3,7], most of the pre-clinical studies mentioned in this review treated animals by intraperitoneal (I.P) injection which in contrast to inhaling induce greater effect in shorter time [9,32]. Furthermore, it is important to take into account that most of the mentioned pre-clinical studies have used specific SCs or pure Δ9-THC for exploring their exclusive effect on a chosen factor [3,9,32]. In contrast to that, evidence from epidemiological data or human studies present information regarding the effects of SC or cannabis products which mostly contain a range of cannabinoids and in some cases additional psychoactive compounds [1,3,7,9,32].

Understanding the effects of cannabinoids on EF has considerable practical utility in the clinical setting. Executive function is essential to an individual’s multiple abilities in daily life [77]. It has been suggested that due to impaired EF, patients may have difficulties in learning new coping behaviors and accordingly increases the likelihood of treatment dropout and poor treatment outcomes [12]. Therefore, the current review emphasizes the need of attention by the clinician regarding cognitive abilities of patients who suffer from cannabinoid abuse. In case of cognitive impairments, an alternative unique therapeutic method should be considered such as behavioral therapy [150] or introducing the patient with cognitive rehabilitation strategies [12]. This may be crucial, especially in cases of patients who are heavy cannabinoid users, or young patients who used cannabinoids in early age for persistent periods. 

## Figures and Tables

**Table 1 brainsci-08-00040-t001:** Common clinical adverse effects induced after consuming synthetic cannabinoids.

Type of Effects	Symptoms
Psychosis	Recurrent psychosis episodes [9,14,15,16].
Agitation	Last for several hours after intoxication of SC [16,17,18].
Affect disturbance	Severe anxiety symptoms and panic attacks shortly after consuming SC [14,17,18,19,20].
Cognitive alterations	Impairment in memory and attention deficits [14,20,21,22]
Cardiovascular effects	Both tachycardia, tachyarrhythmia and cardiotoxicity were reported after exposure to SC [14,23].
Gastrointestinal effects	Nausea, vomiting and diarrhea after severe exposure to SC [14,24,25].

**Table 2 brainsci-08-00040-t002:** Pre-clinical rodent studies of the effects of cannabinoid-agonists on executive function

Animals	Cannabinoids Tested	Main Findings	Reference
Male Long–Evans rats	WIN55,212-2 and Δ9-THC	Dose-related attention impairments afteracute exposure to cannabinoid CB_1_ receptor agonist. Impairments were reduced after treatment with CB_1_ antagonist.	[26,27]
Male Sprague–Dawley rats	Δ9-THC	Decreased performance on a divided attention tasklasts for 2 weeks after chronic administration withcannabinoid CB_1_ receptor agonist.	[28]
Male Sprague–Dawley rats	AM-4054	Decreased sustained attention after acute treatmentwith a cannabinoid CB_1_ receptor agonist.Impairments were associated with task demands.	[14]
Male Sprague–Dawley rats	Δ9-THC	Impairments of visual attention on an operant signaldetection task after acute treatment with cannabinoid CB_1_ receptor agonist.	[29]
Male Sprague–Dawley rats	WIN55,212-2	Deficits of working memory after chronic treatmentwith a cannabinoid CB_1_ receptor agonist.	[30]
Female Long–Evans rats	Δ9-THC	Repeated administration with cannabinoid CB_1_ receptoragonist in adolescence induced persistent impairment of working memory.	[31]
Male Sprague–Dawley rats	WIN55,212-2	Acute injection of cannabinoid CB_1_ receptor agonist in late-adolescence period induced temporary impairment of short-term memory. Chronic treatment with cannabinoid CB_1_ receptor agonist impair short-term memory for several weeks after the last administration.	[32]
Male Sprague–Dawley rats	Δ9-THC	Acute exposure to a cannabinoid CB_1_ receptor agonistinduced working memory impairments	[33,34]
Male Sprague–Dawley rats, Lister rats and C57B16 mice	Δ9-THC	Working memory impairments were induced afterchronic treatment with a cannabinoid CB_1_ receptor agonist.	[35,36]
Wild-type and CB_1_ receptor knockout mice	JWH-081	Acute treatment with cannabinoid CB_1_ receptor agonist induced short-term memory deficits in wild-type mice but not in knockout mice.	[37]
Male Long–Evans rats	HU-210	Acute treatment with a cannabinoid CB_1_ receptor agonistinduced working memory deficits.	[38,39]
Male C57B1/6 mice	Δ9-THC	Acute injection of Δ9-THC disrupted performance of the working memory task, impairments were reversed by SR1417161A.	[40]
Male Wistar Rats	Δ9-THC	Acute administration induced set-shifting impairments24 h after treatment.	[41]
Male albino Wistar rats	Δ9-THC	Acute treatment with a cannabinoid CB_1_ receptor agonistinduced short-term memory deficits, impairments were attenuated after treatment with cannabinoid CB_1_ antagonist.	[42]
Male ICR (CD-1) mice	JWH-018, JWH-018-Cl, JWH-018-Br and Δ9-THC	SCs dose-dependently impaired short- term memory. Their effects resulted more potent respect to that evoked by ∆9-THC.	[43]
Male Long–Evans rats	JWH-018	Chronic exposer to cannabinoid CB_1_ receptor agonist induced spatial learning and short-term memory alterations well after the drugs exposure period.	[44]
Male Lister Hooded and Wistar rats	CP55,940	Acute administration of cannabinoid CB1 receptor agonist impaired short-term memory in both strains, yet, no long-term effects were observed.	[45]
Male Long–Evans rats	Δ9-THC	Acute treatment with a cannabinoid CB_1_ receptor agonistinduced reversal learning deficits while set-shifting ability has maintained	[46]
Male Long–Evans rats	HU-210	Administration of the cannabinoid CB_1_ receptor agonistelicited dose-dependent disruptive effects on set-shiftingperformance. Impairments were diminished afteradministration of the CB_1_ antagonist AM251.	[47]
Male Albino Wistar rats	AB-PINACA or AB-FUBINACA compere with Δ9-THC	Two weeks after repeated administration of cannabinoid-agonist short-term memory impairments were observed, in SCs groups the impairments were greater and last for longer time.	[48]
Female and Male Sprague–Dawley rats	WIN55,212-2	Self-administration of SCs in low dosages during adolescence period improve or did not induce permanent memory impairments, while treatments of high dosages of SCs in adolescence period induced permanent short-term memory impairments.	[49,50]

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
