# Peer review of "The Effects of Cannabinoids on Executive Functions: Evidence from Cannabis and Synthetic Cannabinoids—A Systematic Review"

_brainsci, 2018, doi:10.3390/brainsci8030040_

Reviewer 1 Report

The manuscript is written well. However Table 2 lacks heading as well it lacks essential recent findings on synthetic cannabinoids such as JWH-081, XLR-11, JWH-018, etc. There are many memory-related studies performed with SC in mice that are not referenced in this review. The review has to be revised thoroughly citing all the work on pre-clinical studies. 

Response- we have revised the pre-clinical studies which we have mentioned in the review and we have added recent pre-clinical rodent studies which examined the effects of SC on short-term memory. In addition, we have  revised table 2. The reviewer has noted there are many memory-related pre-clinical studies performed with cannabinoids in mice, yet, we intently limited the current review to present studies which explore the effects of cannabinoids on executive functions only (short-term/working memory, cognitive flexibility, and attention as defined by Diamond, 2013). Therefore, we did not cite studies which investigated further cognitive domains such as long-term memory, emotive memory, etc. 

Reviewer 2 Report

In the manuscript by Cohen and Weinstein, the authors deal with the most popular illicit drug, cannabis. The authors focus their attention on the effects of cannabis and new types of drugs containing synthetic cannabinoids that appear to have the same adverse effects seen for cannabis. They report that pre-clinical as well as clinical studies demonstrated an association between synthetic cannabinoids consumptions and cognitive functions impairment in general and executive function specifically. The authors emphasize the need of attention by the clinician regarding cognitive impairments of patients who suffer from cannabinoid abuse.

In general, the structure of the proposed review is presented clearly and the analysis is accurate and precise; the tables are instrumental for the content of the review. Therefore, in my opinion, it is essential to encourage the publication of such insights, because these activities improve knowledge within the scientific community and keep vigilant on this public health problem.

In conclusion, I highly recommend acceptance of this manuscript for publication in Brain Sciences.

Author Response

We are very grateful for the reviewer's positive evaluation of the manuscript. We have hired an English native speaker to edit and correct and spelling or errors of style.

Reviewer 3 Report

The aim of the systematic review by Cohen & Weinstein was to give an overview of the evidence for effects of cannabinoids on executive functions. The authors included preclinical and clinical studies for organic cannabinoids and synthetic cannabinoids (SC). Acute as well as chronic effects were reviewed. This is a challenging endeavor.

Major comments

-       The strength and at the same time the main problem of the manuscript is the accumulation of a huge number of studies (142 references). The manuscript can be improved by better structuring of the material and critical interpretation of the results.

Minor comments

-       Do not sufficiently address the pharmacology of SC. Mention higher affinity but do not relate those comments to physiologic effects. Do not discuss the metabolism of SC which could give insight into the longevity of its effects. 

-       Psychological tests are incompletely explained.

-       Do not address how the age at the beginning of use or frequency of use might have impact on executive functioning. Should have also addressed variations in methods of administration.

-       Do not address limitations of using pre-clinical studies (e.g., method of administration: i.p.).

-       Do not talk about how SC are packaged and how this might have deleterious effects (more than one SC in Spice, K2) Do not talk about the additional chemicals that might also be associated with CNS effects (e.g. seizures).

-       Some references questionable, e.g. #42.

-       Dosages reported incorrectly (mg, not mg/kg).

-       Spelling and grammar mistakes throughout (use of preforming vs performing- lines 182, lines 191). Run on sentences (line 191 and more).

Author Response

Reviewer 3

The aim of the systematic review by Cohen & Weinstein was to give an overview of the evidence for effects of cannabinoids on executive functions. The authors included preclinical and clinical studies for organic cannabinoids and synthetic cannabinoids (SC). Acute as well as chronic effects were reviewed. This is a challenging endeavor.

Major comments

-       The strength and at the same time the main problem of the manuscript is the accumulation of a huge number of studies (142 references). The manuscript can be improved by better structuring of the material and critical interpretation of the results.

Response- we realize that this is an extensive review. We have therefore divided it into sections with theoretical background, pharmacology, different types of cognitive processes such as attention, executive function, working memory, cognitive flexibility and we have divided the sections into pre-clinical and clinical studies to make the review coherent. We have also added a paragraph describing the limitation of the current review.

Minor comments

 -       Do not sufficiently address the pharmacology of SC. Mention higher affinity but do not relate those comments to physiologic effects. Do not discuss the metabolism of SC which could give insight into the longevity of its effects. 

Response- we have now added wider explanation regarding the psychoactive compounds of SC drugs focused on the different classes of current known SCs were found in SC products and described their characteristics.

-       Psychological tests are incompletely explained.

Response- we have added further explanations of the psychological tests of the WM and cognitive flexibility tasks.

-       Do not address how the age at the beginning of use or frequency of use might have impact on executive functioning. Should have also addressed variations in methods of administration.

Response- we have added the effect of early consumption of cannabinoids in relations with the ECS function (Lines 453-463). In addition, we have now discussed the role of administration methods in the conclusions.

-       Do not address limitations of using pre-clinical studies (e.g., method of administration: i.p.).

Response- limitations are now added to the conclusions part see lines 472-482.

-       Do not talk about how SC are packaged and how this might have deleterious effects (more than one SC in Spice, K2) Do not talk about the additional chemicals that might also be associated with CNS effects (e.g. seizures).

Response- The presence of additional chemicals in SC products were added (lines 115-119), in addition see lines for the diversity in packaging of these drugs and their influence (lines 77-87).

-       Some references questionable, e.g. #42.

Response- References list is now revised and corrected

-       Dosages reported incorrectly (mg, not mg/kg).

Response- dosages report is now revised and corrected.

-       Spelling and grammar mistakes throughout (use of preforming vs performing- lines 182, lines 191). Run on sentences (line 191 and more).

Response- we have now hired a native English speakers who revised and edited the text for spelling and grammar mistakes.

Round  2

Reviewer 1 Report

Authors somewhat addressed my previous concerns. 

Reviewer 3 Report

The authors revised the manuscript according to the comments of the reviewer.